# Overexpression of *bmp4*, *dazl*, *nanos3* and *sycp2* in Hu Sheep Leydig Cells Using CRISPR/dcas9 System Promoted Male Germ Cell Related Gene Expression

**DOI:** 10.3390/biology11020289

**Published:** 2022-02-11

**Authors:** Hua Yang, Mingtian Deng, Wenli Lv, Zongyou Wei, Yu Cai, Peiyong Cheng, Feng Wang, Yanli Zhang

**Affiliations:** 1Institute of Sheep and Goat Science, Nanjing Agricultural University, Nanjing 210095, China; 15062275837@163.com (H.Y.); mtdeng@njau.edu.cn (M.D.); 15118104@njau.edu.cn (W.L.); 2021205019@njau.edu.cn (Y.C.); 2020105038@njau.edu.cn (P.C.); caeet@njau.edu.cn (F.W.); 2Taicang Agricultural and Rural Science & Technology Service Center, Graduate Workstation, Taicang 215400, China; tcnltz@163.com

**Keywords:** CRISPR/dcas9, Leydig cells, germ cells, target gene activation, sheep

## Abstract

**Simple Summary:**

Male germ cell development plays a crucial role in male reproduction, and gene expression also presents an essential regulatory role in its development. Many studies have been devoted to the induction and differentiation of pluripotent stem cells into germ cells in vitro. However, the culture system for pluripotent stem cells from domestic animals is not stable, especially in sheep. Our study attempted to transdifferentiate sheep somatic cells into germ cells in vitro by the overexpression of key germ cell related genes, with the aim of perfecting the construction of germ cell research models in vitro. Therefore, we explored the expression pattern of four crucial genes, *bmp4*, *dazl*, *nanos3* and *sycp2*, in Hu sheep testicular development, and investigated the potential efficiency of overexpression of the four candidate genes using the CRISPR/dcas9 system in Leydig cells. We revealed that the overexpression of *bmp4*, *dazl*, *nanos3* and *sycp2* can promote the expression of male germ cell related genes. To the best of our knowledge, this is the first study to construct an overexpression induction system using CRISPR/dcas9 technology, and to induce sheep somatic cells into germ cells in vitro.

**Abstract:**

Male germ cells directly affect the reproduction of males; however, their accurate isolation and culture in vitro is extremely challenging, hindering the study of germ cell development and function. CRISPR/dcas9, as an efficient gene reprogramming system, has been verified to promote the transdifferentiation of pluripotent stem cells into male germ cells by editing target genes. In our research, we explored the expression pattern of the germ cell related genes *bmp4*, *dazl,*
*nanos3* and *sycp2* in Hu sheep testicular development and constructed the overexpression model using the CRISPR/dcas9 system. The results indicated that four genes showed more expression in testis tissue than in other tissues, and that *bmp4*, *dazl* and *sycp2* present higher expression levels in nine-month-old sheep testes than in three-month-olds, while *nanos3* expressed the opposite trend (*p* < 0.05). In addition, the expression of four potential genes in spermatogenic cells was slightly different, but they were all expressed in sheep Leydig cells. To verify the potential roles of the four genes in the process of inducing differentiation of male germ cells, we performed cell transfection in vitro. We found that the expression of the germ cell related genes Prdm1, Prdm14, Mvh and Sox17 were significantly increased after the overexpression of the four genes in Leydig cells, and the co-transfection effect was the most significant (*p* < 0.05). Our results illustrate the crucial functions of *bmp4*, *dazl*, *nanos3* and *sycp2* in Hu sheep testis development and verified the effectiveness of the overexpression model that was constructed using the CRISPR/dcas9 system, which provided a basis for further male germ cell differentiation in vitro.

## 1. Introduction

Infertility is a persistent reproductive health problem worldwide, and approximately half of cases are attributable to males [1,2]. Spermatogenesis is a crucial process in male reproduction, including testicular somatic cell development and germ cell differentiation. Male germ cell development participates in the complex process of mitosis and meiosis, and male germ cells at different developmental stages are difficult to isolate accurately. Disorders of germ cell development can cause many male diseases, which in turn cause male fertility problems [3]. It is important to investigate male germ cell development and regulatory mechanisms.

In many studies, stem cell transdifferentiation is widely used to study the development of germ cells [4,5], including cytokine induction [6], somatic cell co-culture [7] and overexpression of key germ cell genes [8]. With the evolution of single-cell sequencing technology, more studies have focused on the identification of the crucial genes in different cell types of the testes [9,10]. In our previous research, we also explored the cell type clustering and marker gene screening of adult Hu sheep testes [11]. In addition, numerous genes have been demonstrated to be pivotal during testicular development and are involved in the complex process of meiosis. *Bmp4* has been reported to be involved in the formation and development of primordial germ cells (PGCs). *Bmp4* can enhance the methylation of H3K4me2 by activating the WNT pathway to ensure the formation of PGCs [12]. It was also reported that lncRNA *bmp4* promoted *bmp4* expression and regulated PGC production [13]. The expressions of *dazl* and *nanos3* were closely related to sheep testicular development [14,15,16]. A previous study indicated that *sycp2*, as an important factor in transcriptional regulation, can interact with lncRNA and potentially regulate spermatogenesis [17]. In addition, many key spermatogenesis related genes have been proven to play pivotal regulatory roles in germ cell differentiation. Genes such as *stra8*, *boule* and *dazl* were overexpressed in goat bone mesenchymal stem cells (BMSCs), which induce goat male germ cells by RNA transfection [18,19]. Overexpressed *cd61* also promoted human umbilical cord mesenchymal stem cell (hUC-MSC) differentiation into male germ-like cells [20]. *Tfcp2l1* was proven to play a key regulatory role in the differentiation of human pluripotent stem cells into primordial germ cell-like cells (PGCLCs) by plasmid transfection [21]. In human induced pluripotent stem cells, the overexpression of *dazl*, *boule* and *daz* effectively promotes meiosis progression and haploid formation [22,23]. The activation of *stra8* and *prm1* in embryonic stem cells induced male germline stem cell differentiation [24]. It was also indicated that Nanog alone promoted the differentiation process of PGCLCs, which showed germline-specific epigenetic modification [25]. Moreover, *bmp4*, as a key factor for inducing pluripotent stem cells to differentiate into male germ cells [26,27,28], is also essential for the regulation of testicular development and spermatogenesis [29,30]. Previous research also certified that *nanos3* played roles in maintaining the number of germ cells and the process of meiosis [31]. Based on our research, *sycp2* exhibited sequential expression in the process of male germ cells [11]. Furthermore, the CRISPR/dcas9 system is an effective means of cellular reprogramming [32,33]. It was reported that human foreskin fibroblasts were reprogrammed into Leydig-like cells based on activating *nr5a1*, *gata4* and *dmrt1* using the CRISPR/dCas9 SAM system [34].

Since Hu sheep are a traditional breed across China, it is important to study their male germ cell development. In the present study, we first explored the expression pattern and location of four crucial genes, *bmp4*, *dazl*, *nanos3* and *sycp2*, in Hu sheep testicular development. Next, the activation system of four genes was constructed using CRISPR/dcas9 and the effectiveness was validated in sheep Leydig cells. Therefore, this study may be valuable for male sheep breeding.

## 2. Materials and Methods

The experiments performed in our study were approved by the Institutional Animal Care and Use Committees at Nanjing Agricultural University, China (Approval ID: SYXK2011-0036).

### 2.1. Animals and Sample Collection

Six healthy male three-month-old and three nine-month-old Hu sheep were selected and slaughtered for testes and internal organ tissue (heart, liver, kidney, rumen, jejunum, ileum, hypothalamus, pituitary and ovary) collection. The tissues were immediately transferred into a −80 °C environment for RNA and protein extraction. Samples trimmed from testes tissue were also fixed in 4% paraformaldehyde for 48 h, and then were dehydrated thorough graded alcohol and clarified in xylene. Then, the tissues were embedded in paraffin, and 5 μm-thick tissue sections were cut for immunostaining.

### 2.2. Total RNA Extraction and qPCR

The total RNA of collected tissues was extracted using TRIzol reagent (Invitrogen Life Technologies, Carlsbad, California, USA) and the concentration and quality were detected using a NanoDrop instrument (NanoDrop Technologies, Wilmington, DE, USA). The qualified RNA was reverse-transcribed into a cDNA template, and further qPCR was performed. The primers of the genes were designed based on the NCBI database and synthesized by Nanjing Tsingke Biotechnology Company (Table 1). The qPCR experiment was carried out following the SYBR qPCR master mix reagent instructions (Vazyme, Nanjing, China). The results of qPCR were analyzed using the 2^−ΔΔt^ method.

### 2.3. Protein Isolation and Western Blot

The protein of the samples was collected using a RIPA lysis solution that contained 1% PMSF, and the concentration was analyzed using a BCA kit (Beyotime Biotechnology Company, Shanghai, China). Denatured protein samples were electrophoresed using 12% bis-tris gel, then transferred to a polyvinylidene difluoride membrane (PVDF). After blocking the PVDF using 5% skimmed milk powder, the brands were incubated with primer antibodies for 16 h at 4 °C. The antibody usage information is shown in Table 2. Following secondary antibody incubation for 2 h, the bands were imaged in the chemiluminescence detection system, and the grayscale value was calculated using ImageJ software.

### 2.4. Tissue Immunohistochemistry and Cellular Immunofluorescence

Sheep testes tissue sections of 5 µM were cut and immunohistochemistry was performed. In addition, sheep Leydig cells were seeded in plates and fixed with 4% paraformaldehyde for immunofluorescence. After cell permeabilization and blocking were performed with TritonX-100 and BSA, respectively, the antibody incubation was carried out. The primary antibodies of Bmp4, Dazl, Nanos3 and Sycp2 were incubated in a dilution of 1:100 at 4 °C overnight, excluding the control group. After washing, the secondary antibody incubation was performed, including the HRP-labeled goat anti-rabbit IgG (H + L) (Beyotime Biotechnology Company, Shanghai, China; A0208, 1:200 dilution) for immunohistochemistry, and CoraLite594–conjugated goat anti-rabbit IgG (H + L) (Proteintech, Wuhan, China; SA00013-4, 1:200 dilution) for immunofluorescence. The positive reaction of tissues sections was detected using a DAB kit (Boster, CA, USA, AR1022). In addition, the nuclei of the tissue sections and cells were stained using hematoxylin and DAPI, respectively. Finally, the tissue sections and cells were imaged under the microscope.

### 2.5. Plasmid Construction and Synthesis of bmp4, dazl, nanos3 and sycp2

To increase gene expression, the sgRNAs of *bmp4*, *dazl*, *nanos3* and *sycp2* were designed on the website (http://chopchop.cbu.uib.no/, 17 Janunary 2021) and listed in Table 3. The pLenti-U6-sgRNA-PGK-Neo backbone, dCas9 Synergistic Activation Mediator Lentivector (K015) and CRISPR Scrambled sgRNA CRISPR Lentiviral Vector (K018) were purchased from Applied Biological Materials Inc. (abm). All of these constructs were confirmed by sequencing.

### 2.6. Cell Isolation Culture and Transfection

Leydig cells from the ram testes were isolated and the method used was as described in previous research [35]. Briefly, fresh and intact testis tissues were brought back to the lab within 1 h, the testis tissues were chopped into 1 mm^2^ pieces, and then digested with 1 mg/mL concentration of collagenase IV for 30 min. Then, the cell suspension was filtered with a 70 μm filter. The culture medium contained 90% DMEM/F12 (Gibco Life Technologies, Carlsbad, CA, USA), 10% FBS (Gibco Life Technologies) and 2 mM L-glutamine, 100 U/mL penicillin, and 100 μg/mL streptomycin (Gibco Life Technologies) was used for cell culture at 37 °C and 5% CO_2_ atmosphere. The cells were seeded in the dish for 1 h, then non-adherent spermatogenic cells were removed. The cells were confluent to 90% density, cells were digested and passaged for further experiment. The cells were seeded in six-well plates and the plasmids of four genes (the total amount of DNA was 4 ug) were transfected into Leydig cells with 70–80% density following the Lipofectamine™3000 transfection reagent instruction. After 48 h and 72 h, the RNA and the protein of cells were extracted, respectively, and further used for the qPCR and Western blot experiments.

### 2.7. Data Analysis

All experiments were performed independently at least three times and the data were expressed as mean ± standard error of mean (SEM). The different letters such as a, b, c and *** represent a significant and extremely significant difference between groups (*p* < 0.05 and *p* < 0.001, respectively) in qPCR experiment.

## 3. Results

### 3.1. The Expression of bmp4, dazl, nanos3 and sycp2 in Sheep Testicular Development

To detect the expression levels of *bmp4*, *dazl*, *nanos3* and *sycp2* in various tissues of sheep, a qPCR experiment was carried out. We found that *bmp4*, *dazl* and *nanos3* presented the highest expression level in testis tissue (Figure 1, *p* < 0.05). *Sycp2* showed moderate expression in whole tissues. The immature and mature sheep testis tissues were used for qPCR and Western blot analysis to reveal the expression patterns of *bmp4*, *dazl*, *nanos3* and *sycp2* during sheep testicular development. As shown in Figure 2, *bmp4*, *dazl* and *sycp2* were more extremely increased in the 9M group than in the 3M group (*p* < 0.001) and *nanos3* presented the opposite trend. Thus, these four genes potentially play roles in sheep testicular development.

### 3.2. Location Analysis of bmp4, dazl, nanos3 and sycp2 in Sheep Testis

For identifying the location of Bmp4, Dazl, Nanos3 and Sycp2 in the sheep testicular development process, an IHC experiment was performed. As shown in Figure 3, Bmp4 was mainly expressed in Leydig cells of the 3M group and abundantly expressed in the spermatocytes and round spermatids of the 9M group. Dazl showed slight expression in 3M testes, but it was strongly expressed in the spermatogonia and spermatocytes of the 9M testes. Nanos3 was expressed in the whole cell types of testes at different stages. Sycp2 expression in the 3M group was similar to that of BMP4, and it was mainly expressed in spermatocytes and spermatids in the 9M group. In total, four genes were expressed in sheep testis Leydig cells.

### 3.3. The Expression of bmp4, dazl, nanos3 and sycp2 in Sheep Leydig Cells

Sheep Leydig cells were isolated and cultured. As shown in Appendix A, the Leydig cell marker genes Hsd3b and Cyp17a1 were positively expressed in the cytoplasm of the isolated cells, and the expression level of other cell type marker genes were significant lower than Leydig cell markers which confirmed that the isolated cells were of high purity. Figure 4 shows that Bmp4 and Nanos3 were mainly expressed in the cytoplasm, while Dazl and Sycp2 were both expressed in the nucleus and the cytoplasm. These results potentially verified the roles of the four genes in sheep testicular cells.

### 3.4. sgRNA Transfection and Screening of bmp4, dazl, nanos3 and sycp2

In order to effectively construct overexpression vectors for *bmp4*, *dazl*, *nanos3* and *sycp2*, we first isolated Hu sheep fibroblasts from ears, then the dCas9 Synergistic Activation Mediator Lentivector and the CRISPR Scrambled sgRNA CRISPR Lentiviral Vector of the four genes were transfected. Figure 5 shows that the sgRNA2, sgRNA2, sgRNA1 and sgRNA1 of *bmp4*, *dazl*, *nanos3* and *sycp2*, respectively, were the most effective in their overexpression levels. In addition, we detected the transfection efficiency of different combinations of sgRNA and found that single sgRNA transfection was more efficient than mixed transfection. Thus, the most effective sgRNA of each gene was selected for further function validation in sheep Leydig cells.

### 3.5. In Vitro Overexpression of bmp4, dazl, nanos3 and sycp2 in Sheep Leydig Cells

Based on the sgRNA screening, we conducted single gene transfection (C, B, D, N, S) and mixed gene transfection (+) in Leydig cells. The Western blot experiments indicated that, respectively, overexpression of *bmp4*, *dazl*, *nanos3* and *sycp2* can improve the expression of these four genes (Appendix A, *p* < 0.05). However, overexpression of *bmp4*, *dazl* and *nanos3*, respectively, did not increase the expression of Prdm1 and Prdm14. In B, D, N, S groups, the protein levels of Mvh and Sox17 were significantly increased. Moreover, the mixed group (+) significantly improved the expression of the male germ cell related genes, Prdm1, Prdm14, Mvh and Sox17, and the levels were higher than B, D, N, S groups, except Sox17 (Figure 6, *p* < 0.05). These results demonstrated that the mixed transfection of four genes was the most effective.

## 4. Discussion

In the study of male reproduction, germ cell development is crucial and essential. However, the separation of spermatogenic cells at different stages in the process of spermatogenesis remains difficult. Research focused on the differentiation of male germ cells induced by other cells through gene editing methods in vitro has been increasing [36]. Previous studies have revealed multiple functionally redundant genes in mouse testicular development using CRISPR/cas9 genome editing technology [37,38]. Nevertheless, few studies have investigated male sheep germ cell development. In our study, we selected four potential genes, *bmp4*, *dazl*, *nanos3* and *sycp2*, to explore their expression pattern in sheep testicular development and construct overexpression vectors for sheep germ cell induction.

To explore gene regulation in the spermatogenesis process, more studies have revealed gene expression profiles in the spermatogenesis process using single-cell sequencing methods [39,40,41]. Previously, we have demonstrated several genes that potentially affect the spermatogenesis process in Hu sheep, and screened out several genes that are expressed sequentially in spermatogenesis [11]. During testicular development, *bmp4*, as a vital transcription factor, was secreted from Sertoli cells in the prepuberal stage and then produced by spermatogonia in the adult stage, ensuring they were involved in the proliferation and differentiation of spermatogonia [42,43]. However, we found that *bmp4* was expressed in Leydig cells during puberty and in spermatogenic cells during sexual maturity in sheep. *Dazl*, a conserved gene related to meiosis [44], showed higher expression levels in adult testes than in the immature testis tissues of goats and sheep [14,45], which was consistent with our results. *Dazl* was located in sheep Leydig cells and regulated sheep spermatogenesis in the post-pubertal stage [15]. In our previous research, *Nanos3* participated in Hu sheep germ cell development by affecting the synthesis and secretion of testosterone [16]. In embryonic development and the PGC differentiation process, *nanos3* displayed crucial functions [46,47] and also maintained the undifferentiated state of spermatogonia [48]. Based on Hu sheep testicular single cell transcriptome analysis [11], *sycp2* manifested potential roles in spermatogenesis. Because it is the major component of the synaptonemal complex, it potentially affects the meiosis process in spermatogenesis [49]. However, little attention has focused on its function in testicular development, and we indicated the expression pattern and location of *sycp2* in sheep testis tissues. Our results explained the expression patterns of the four candidate genes in Hu sheep testicular development, and we conducted a localization analysis in Leydig cells, providing the basis for further vector construction and verification experiments.

The study of male spermatogenesis disorders always involves the separation of spermatogenic cells. Many studies have focused on the induced differentiation of male germ cells from pluripotent stem cells using the CRISPR/cas9 system [50]. The CRISPR/cas9 system has the advantage of high targeting efficiency, low off-target effects and low technical difficulty, and it also has been successfully used for the treatment of clinical diseases [51]. It has been reported that both *c1eis* and *stra8* gene deletion mediated by CRISPR/cas9 inhibit the differentiation of embryonic stem cells into male germ cells [52,53]. The CRISPR/dcas9 system can effectively activate gene expression through the combination of the *vp64* domain and transcriptionally activated dCas9 protein [54,55]. In our study, we used this activation system to construct a multi-gene transcription activation vectors for exploring the importance of gene induction in the differentiation of male germ cells in vitro.

In the present study, we first detected the expression changes and localization analysis of *bmp4*, *dazl*, *nanos3* and *sycp2* in Hu sheep testis development, and isolated testicular Leydig cells in vitro. We then verified the efficacious overexpression of the four genes in vitro by constructing overexpression vectors using the CRISPR/cas9 activation system. Our results will be helpful for further perfecting the differentiation of male germ cells in vitro and providing new insights for livestock breeding.

## 5. Conclusions

In conclusion, our study focused on the overexpression system construction in sheep testicular somatic cells using CRISPR/cas9 genome editing technology. The results indicated that the candidate genes *bmp4*, *dazl*, *nanos3* and *sycp2* presented different expression changes and locations during testicular development. The expression of germ cell related genes was promoted after the overexpression of the four genes using the CRISPR/cas9 system. The present results proved the feasibility of the overexpression system construction using CRISPR/dcas9 in sheep Leydig cells.

## Figures and Tables

**Figure 1 biology-11-00289-f001:**
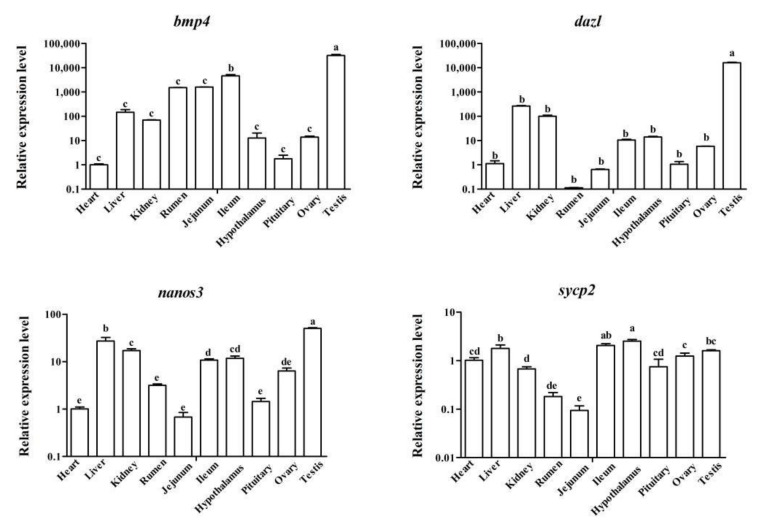
qPCR analysis of mRNA expression of *bmp4*, *dazl*, *nanos3* and *sycp2* in different tissues of sheep. The expression of *bmp4*, *dazl*, *nanos3* and *sycp2* in whole sheep tissues was analyzed using qPCR. The results were expressed relative to the heart tissue as mean values ± the SEM. Different letters represent significant differences between groups, *p* < 0.05.

**Figure 2 biology-11-00289-f002:**
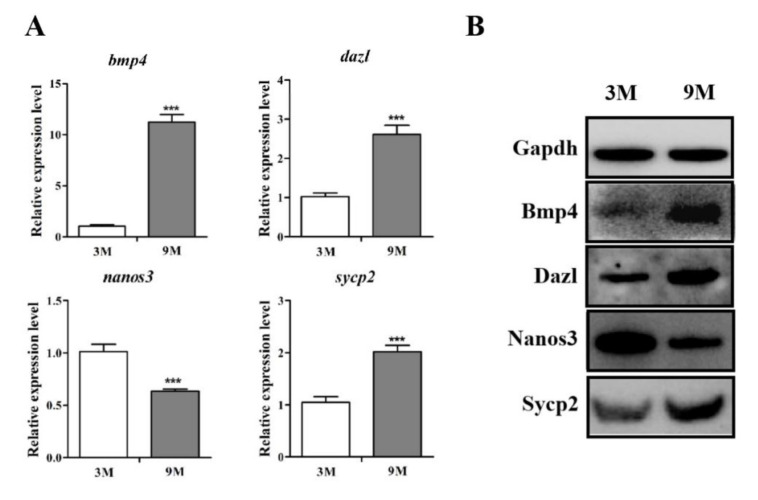
The expression levels of *bmp4,*
*dazl,*
*nanos3* and *sycp2* in sheep testicular development. (**A**,**B**) represent the mRNA and protein expression levels of *bmp4,*
*dazl*, *nanos3* and *sycp2* in the 3M and 9M group using qPCR and Western blot, respectively. qPCR results were expressed relative to the 3M group as mean values ± the SEM. *** indicates extremely significant differences between groups (*p* < 0.001); 3M and 9M refer to 3-month-old and 9-month-old testis tissues of sheep, respectively.

**Figure 3 biology-11-00289-f003:**
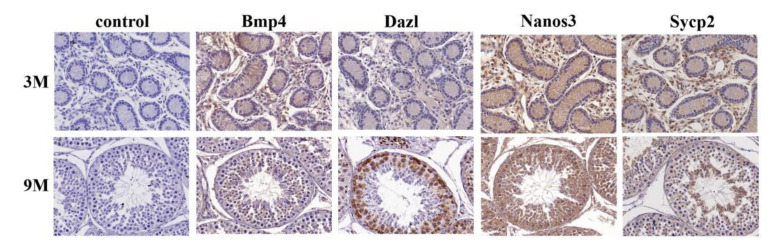
The expression location of Bmp4, Dazl, Nanos3 and Sycp2 in 3M and 9M sheep testis tissues. Immunohistochemistry was performed to detect the expression location of the four genes in sheep testis tissues. The staining results were observed under a microscope at a magnification of 40×. Control presented the negative staining; Bmp4, Dazl, Nanos3 and Sycp2 presented the positive staining. Blue and brown represented nuclear and gene positive staining, respectively. The terms 3M and 9M refer to the 3-month-old and 9-month-old testis tissues of sheep, respectively.

**Figure 4 biology-11-00289-f004:**
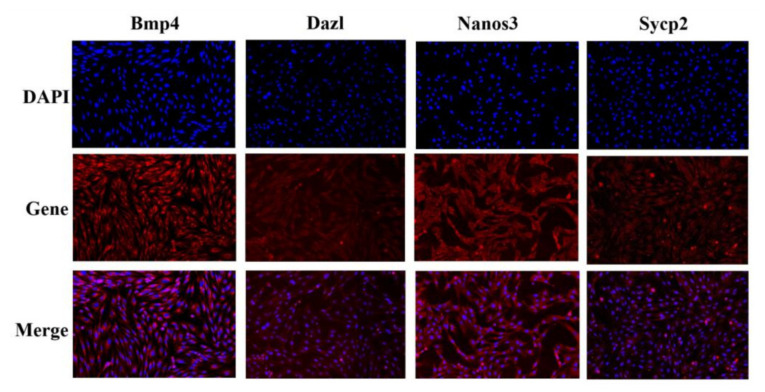
The expression location of Bmp4, Dazl, Nanos3 and Sycp2 in sheep Leydig cells. To locate the expression of Bmp4, Dazl, Nanos3 and Sycp2 in sheep Leydig cells, immunofluorescence was carried out. DAPI indicated the nuclear staining, and the results were photographed under a laser scanning confocal microscope at a magnification of 40×.

**Figure 5 biology-11-00289-f005:**
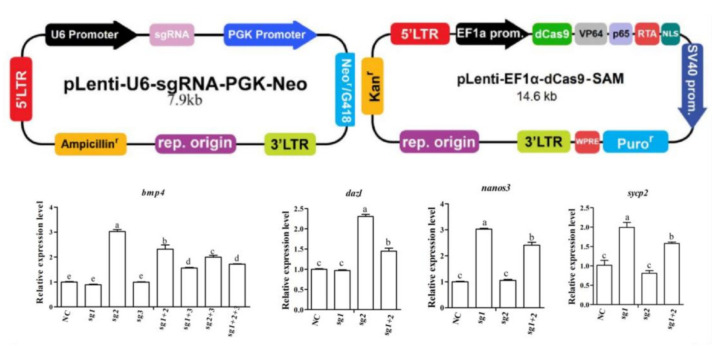
Overexpression of *bmp4*, *dazl*, *nanos3* and *sycp2* in sheep fibroblasts. To overexpress the candidate genes *bmp4*, *dazl*, *nanos3* and *sycp2*, sgRNAs were designed and transfected in sheep ear fibroblasts. The transfection efficiency was proven by the qPCR experiment, and the results were expressed relative to the control group (NC) as mean values ± the SEM. Different letters mean significant differences between groups (*p* < 0.05).

**Figure 6 biology-11-00289-f006:**
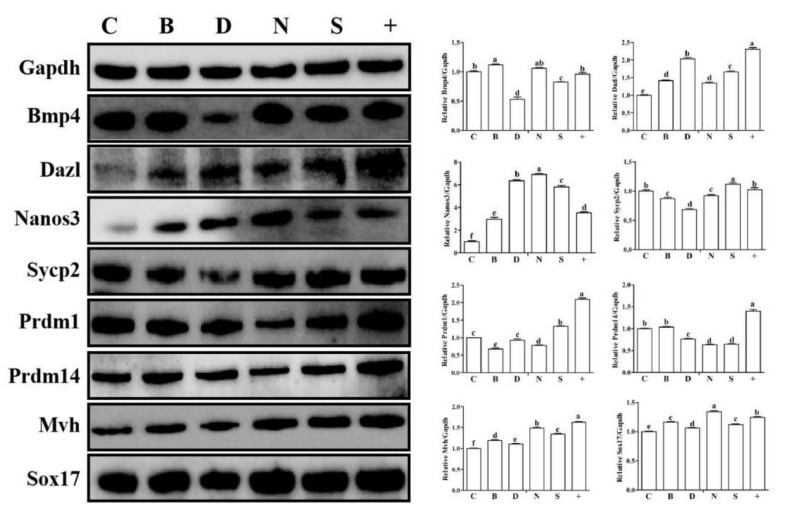
Germ cell related gene expression changes after the overexpression of *bmp4*, *dazl*, *nanos3* and *sycp2* in sheep Leydig cells. Western blot analysis was performed to determine the overexpression efficiency of CRISPR/dcas9 system in vitro. C, B, D, N, S, and + represent the control, *bmp4*, *dazl*, *nanos3*, *sycp2* and mix of four gene groups, respectively. The results were expressed relative to the control group C as mean values ± the SEM. Different letters mean significant differences between groups (*p* < 0.05).

**Table 1 biology-11-00289-t001:** Primer sequences of genes.

Primer Name	Primer Sequence	Product Length/bp
*gapdh*	F: GTCAAGGCAGAGAACGGGAA	
	R: GGTTCACGCCCATCACAAAC	232
*bmp4*	F: CACCTCCATCAGACACGGAC	
	R: CCAGTCATTCCAGCCCACAT	258
*dazl*	F: GAGACTCCAAACTCAGCCGT	
	R: TAGCCTTTGGACACACCAGT	227
*nanos3*	F: GACCTTCAACCTGTGGACAGAC	
	R: CGGTTCTGGCACTGCTTCT	258
*sycp2*	F: TTGGAAAGGGCACAACCAAG	
	R: TGCTCTTCGTGGAAGTCTGG	105

**Table 2 biology-11-00289-t002:** The antibodies information for western blot.

Antibody Name	Catalog No.	Dilution	Source
Gapdh	Proteintech-1E6D9	1:10,000	Mouse
Bmp4	Affinity-DF6461	1:500	Rabbit
Dazl	Abcam-ab215718	1:1000	Rabbit
Nanos3	Abcam-ab70001	1:500	Rabbit
Sycp2	Affinity-DF2578	1:500	Rabbit

**Table 3 biology-11-00289-t003:** sgRNA sequences of four genes.

Gene Name	sgRNA Sequence
*bmp4*	Target 1: TTTCACCGCCCGCCTCGGGG
	Target 2: TATGAGTCACGTGAGCGCAG
	Target 3: CTCTGGATGGCACTACGGAA
*dazl*	Target 1: TCAGCGTCCCGGCCACCCCA
	Target 2: GCCGCGCTTGCCTGTCCTGG
*nanos3*	Target 1: TCTTGGAGGACCGGCTTAGG
	Target 2: GCTCTAGAGGGAGGGTCCTA
*sycp2*	Target 1: AGTAATAAAGACTTTCTCCT
	Target 2: TGAATGAAGTTCCAATTCCA

## Data Availability

Not applicable.

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
