# Peer review of "Overexpression of bmp4, dazl, nanos3 and sycp2 in Hu Sheep Leydig Cells Using CRISPR/dcas9 System Promoted Male Germ Cell Related Gene Expression"

_biology, 2022, doi:10.3390/biology11020289_

Round 1

Reviewer 1 Report

1- In this manuscript, the authors studied the Activation of BMP4, DAZL, NANOS3, and SYCP2 by CRISPR/dcas9 system. Although the results are so interesting and novel, some manuscript should be revised. My suggestions and comments are listed below, in no particular order:

2- The Introduction section of the paper should give the greater context of the project and explain in a paragraph the significance of the BMP4, DAZL, NANOS3, and SYCP2 genes. Please add some recently studied in lines 53 to 58.

3- The discussion section is interesting, but it needs to discuss more and compared with the other studies such as " CRISPR/Cas9-mediated genome editing reveals 30 testis-enriched genes dispensable for male fertility in mice", " CRISPR/Cas9-mediated genome-edited mice reveal 10 testis-enriched genes are dispensable for male fecundity" or other studies. Please add them in lines 226 to 229.

4- Finally, this manuscript was good and I see some grammar or word problems. Please correct them.

Author Response

Response to Reviewer 1 Comments

In this manuscript, the authors studied the Activation of BMP4, DAZL, NANOS3, and SYCP2 by CRISPR/dcas9 system. Although the results are so interesting and novel, some manuscript should be revised. My suggestions and comments are listed below, in no particular order:

  1. The Introduction section of the paper should give the greater context of the project and explain in a paragraph the significance of the BMP4, DAZL, NANOS3, and SYCP2 genes. Please add some recently studied in lines 53 to 58.

Response: Thanks for your comments. We have provided the related research of BMP4, DAZL, NANOS3, and SYCP2 genes in the Introduction section.

  1. The discussion section is interesting, but it needs to discuss more and compared with the other studies such as " CRISPR/Cas9-mediated genome editing reveals 30 testis-enriched genes dispensable for male fertility in mice", " CRISPR/Cas9-mediated genome-edited mice reveal 10 testis-enriched genes are dispensable for male fecundity" or other studies. Please add them in lines 226 to 229.

Response: Thank you for your comments. We have supplemented these two research process related to CRISPR/Cas9 and testicular development, including PMID: 31201419, PMID: 32561905.

  1. Finally, this manuscript was good and I see some grammar or word problems. Please correct them.

Response: Thank you for your comments. We have carefully revised the manuscript, and also corrected many grammatical and spelling errors

Reviewer 2 Report

The manuscript "Activation of BMP4..." describes a study exploring the expression or four genes in sheep testicular development, examining mature and immature cells.

The study is interesting and it could be reconsidered for publication  some text revisions + methods description improving and providing info on control tissue/sections in the immunohistochemical testes performed:

1)  Verbs missing in some sentences, several sentences start with "and" so an extensive English editing of the text is required.

2) Title and all over the text.  Leydig and Sertoli)were two scientists, so their names must be written with capital letter.

3) Affiliations. There are 2 affiliations but it seems that all Authors have the  affiliation "1". Please check and correct, or eliminate one of the two affiliations

4) Simple summary. Is this an important part of the paper, because it explains the importance of the study and contributes to its attractiveness. The present one is quite short and a little bit poor. I suggest to improve it highlighting why the study was decided and than pointing out the importance of the results. Moreover the last words "provides an overexpression system" are unclear for the reader. Please take into account that readers read the Simple summary BEFORE reading the entire paper, and also before the abstract. Simple Summary must be like an "advertising" of your paper highlighting its importance and attracting the attention of the readers, who can be not expert in thefield,  pushing them to read your paper. So, finally, I kindly ask Author to "sell" better" their study formulating a new, more attractive, Simple Summary, including also the general importance of gene expression in the development. Simple summary is not strictly for genetists.

Line 21: delete "which"

Line 30: vilify??? or verify?

Line 45-46: English editing will contribute to improve the sentence

Line 55: delete "genes" after "pivotal"

Line 66: Moreover (no comma) BMP4 (comma)

Line 73: Were not was

Line 75: Being Hu sheep a traditional...

Line 87: haslet?? Which tissue you define haslet?

LIne 89: tissue samples were fixed in paraformaldehide. Please add here also that tehy were processed for histology, dehydrated in graded alcohols, claryfied in xilene and then embedded in paraffin was. From paraffin blocks, 5 micron thick section were obtained for immunostaining

Line 116: which secondary antibody? Counterstaining? Negative and positive controls?? please describe extensively and correctly the immunohistochemical method.  

Line 134: which different letters?

Line 218-267: particular importance of English editing.

Author Response

Response to Reviewer 1 Comments

The manuscript "Activation of BMP4..." describes a study exploring the expression or four genes in sheep testicular development, examining mature and immature cells.

The study is interesting and it could be reconsidered for publication  some text revisions + methods description improving and providing info on control tissue/sections in the immunohistochemical testes performed:

  1. Verbs missing in some sentences, several sentences start with "and" so an extensive English editing of the text is required.

Response: Thank you for your comments. We have carefully revised the manuscript, and also corrected many grammatical and spelling errors

  1. Title and all over the text.  Leydig and Sertoli)were two scientists, so their names must be written with capital letter.

Response: Thank you for your comments and reminder. We have corrected these two words in the full text.

  1. There are 2 affiliations but it seems that all Authors have the  affiliation "1". Please check and correct, or eliminate one of the two affiliations

Response: Thank you for your reminder. We have checked the affiliations and modified this section.

  1. Simple summary. Is this an important part of the paper, because it explains the importance of the study and contributes to its attractiveness. The present one is quite short and a little bit poor. I suggest to improve it highlighting why the study was decided and than pointing out the importance of the results. Moreover the last words "provides an overexpression system" are unclear for the reader. Please take into account that readers read the Simple summary BEFORE reading the entire paper, and also before the abstract. Simple Summary must be like an "advertising" of your paper highlighting its importance and attracting the attention of the readers, who can be not expert in thefield,  pushing them to read your paper. So, finally, I kindly ask Author to "sell" better" their study formulating a new, more attractive, Simple Summary, including also the general importance of gene expression in the development. Simple summary is not strictly for genetists.

Response: Thank you for your comments and suggestions. We have modified this section and summarized the importance of our study.

Line 21: delete "which"

Response: Thank you for your comments. We have deleted the “which” word.

Line 30: vilify??? or verify?

Response: Thank you for your reminder. We have modified this mistake, it is “verify” not “vilify”.

Line 45-46: English editing will contribute to improve the sentence

Response: Thank you for your comments. We have modified this sentence into “Male germ cells development participate in the complex process of mitosis and meiosis, and male germ cells at different developmental stages are difficult to isolate accurately”.

Line 55: delete "genes" after "pivotal"

Response: Thank you for your comments. We have deleted the “genes” word.

Line 66: Moreover (no comma) BMP4 (comma)

Response: Thank you so much for your comments. We have modified this mistake.

Line 73: Were not was

Response: Thank you for your comments. We have modified this word.

Line 75: Being Hu sheep a traditional...

Response: Thank you for your comments. We have modified this sentence.

Line 87: haslet?? Which tissue you define haslet?

Response: Thank you for your comments. We collected the internal organs tissues including heart, liver, kidney, rumen, jejunum, ileum, hypothalamus, pituitary and ovary for qPCR in Figure 1. We have modified the haslet into internal organs tissues and also added the description in method section.

LIne 89: tissue samples were fixed in paraformaldehide. Please add here also that tehy were processed for histology, dehydrated in graded alcohols, claryfied in xilene and then embedded in paraffin was. From paraffin blocks, 5 micron thick section were obtained for immunostaining

Response: Thank you for your comments and suggestions. We have added the detailed steps on the treatment of tissue samples.

Line 116: which secondary antibody? Counterstaining? Negative and positive controls?? please describe extensively and correctly the immunohistochemical method. 

Response: Thank you for your comments. We have modified this section and added the immunohistochemical method.

Line 134: which different letters?

Response: Thank you for your comments. The different letters such as a, b, c represent the significant difference between groups in qPCR experiments, and we have added an explanation about the different letters in method part.

Line 218-267: particular importance of English editing.

Response: Thank you for your comments. We have check the grammar of discussion part and carefully revised the manuscript to ensure the correct grammar and spelling.

Round 2

Reviewer 2 Report

Authors followed the suggestion given and did the corrections.

I suggested that an extensive English editing of the manuscript would improved the article, but it seems me that this was not done.  In any case,  here below other mistyping to correct in the new text (added parts) and small suggestions.

Line 13: presents - not present

Line 17: delete "so as" and substitute with  "with the aim to"

Line 22: acknowledge? Knowledge

Line 101: delete "including" 

Line 103: delete "The" and start sencence with "Samples trimmed from testes tissue were also..."

Line 104 :  delete "the tissues"

Line 104 (at the end): delete "with a" and use "thorough"

Line 105: in xylene

Line 106: delete "a" before "thick"

Line 133: except not expect

Line 133  (at the end): delete "Followed by" and write "After washing, the secondary antibody incubation was performed, including...."

Line 156: data were ("data" is plural)

Line 190: similar to not "similar with"

Line 296: delete "And" - As already remarked, sentences can't start with "And".

Author Response

Response to Reviewer 2 Comments

I suggested that an extensive English editing of the manuscript would improved the article, but it seems me that this was not done.  In any case,  here below other mistyping to correct in the new text (added parts) and small suggestions.

Response: Thank you for your comments. We have carefully revised the manuscript, and we also got the help of language editing in the MDPI system (https://www.mdpi.com/authors/english (https://www.aje.com/) to ensure the correct grammar and spelling.

Line 13: presents - not present

Response: Thank you for your comments. We have corrected this error.

Line 17: delete "so as" and substitute with  "with the aim to"

Response: Thank you for your comments and suggestions. We have deleted "so as" and substitute with "with the aim to".

Line 22: acknowledge? Knowledge

Response: Thank you for your reminder. We have corrected this error.

Line 101: delete "including"

Response: Thank you for your comments. We have deleted "including".

Line 103: delete "The" and start sencence with "Samples trimmed from testes tissue were also..."

Response: Thank you for your comments. We have revised this sentence.

Line 104 :  delete "the tissues"

Response: Thank you for your reminder. We have deleted "the tissues”.

Line 104 (at the end): delete "with a" and use "thorough"

Response: Thank you for your comments. We have deleted "with a" and use "thorough".

Line 105: in xylene

Response: Thank you for your comments. We have corrected the error.

Line 106: delete "a" before "thick"

Response: Thank you so much for your comments. We have deleted "a".

Line 133: except not expect

Response: Thank you for your reminder. We have corrected this word.

Line 133  (at the end): delete "Followed by" and write "After washing, the secondary antibody incubation was performed, including...."

Response: Thank you for your comments. We have modified this sentence.

Line 156: data were ("data" is plural)

Response: Thank you for your comments. We have modified “data was” into “data were”.

Line 190: similar to not "similar with"

Response: Thank you for your comments. We have corrected this error.

Line 296: delete "And" - As already remarked, sentences can't start with "And"

Response: Thank you for your reminder. We have checked the whole manuscript and corrected this error.
